# FPGA Implementation of a Chaotic Map with No Fixed Point

Claudio García-Grimaldo [1], Ciro Fabián Bermudez-Marquez [2], Esteban Tlelo-Cuautle [2]
and Eric Campos-Cantón [1,*]

1   Instituto Potosino de Investigación Científica y Tecnológica A.C. (IPICYT), San Luis Potosí 78216, Mexico
2   Instituto Nacional de Astrofísica, Óptica y Electrónica (INAOE), San Andrés Cholula 72840, Mexico
*   Correspondence: eric.campos@ipicyt.edu.mx

**Abstract:** The employment of chaotic maps in a variety of applications such as cryptosecurity, image encryption schemes, communication schemes, and secure communication has been made possible thanks to their properties of high levels of complexity, ergodicity, and high sensitivity to the initial conditions, mainly. Of considerable interest is the implementation of these dynamical systems in electronic devices such as field programmable gate arrays (**FPGAs**) with the intention of experimentally reproducing their dynamics, leading to exploiting their chaotic properties in real phenomena. In this work, the implementation of a one-dimensional chaotic map that has no fixed points is performed on an FPGA device with the objective of being able to reproduce its chaotic behavior as well as possible. The chaotic behavior of the introduced system is determined by estimating the Lyapunov exponents and its chaotic behavior is also analyzed using bifurcation diagrams. Simulations of the system are realized via Matlab, as well as in C and the very high-speed integrated circuit (**VHSIC**) hardware description language (**VHDL**). Experimental results on FPGA show that they are like those obtained in the simulations; therefore, this chaotic dynamical system could be used as an element in some encryption schemes such as in the generation of cryptographically secure pseudorandom numbers.

**Keywords:** chaotic map; PWL map; map without fixed points; FPGA implementation; Lyapunov exponent; bifurcation diagram

## 1. Introduction

Chaotic dynamical systems have proven to be very useful in several areas of technology, in biological and physical sciences, as well as in different branches of engineering, mainly because of their ergodicity properties, sensitivity to initial conditions, and positive entropy [1–3]. Some dynamical systems, whether defined in continuous or discrete time, have been implemented through electronic circuits and have been given different uses. For example, the implementation of such chaotic systems has been used in image encryption schemes, secure communication systems, etc., and has been done through the use of different electronic devices such as field programmable analog arrays (**FPAAs**) [4], the platform STM32 [5], and FPGAs [6]. These different devices have some advantages and disadvantages, for example, in an FPAA device the processing time is less compared to digital devices because it does not require discretization and the blocks are already established. However, it presents a lower accuracy, the design is closed, and also for digital applications it needs an analog-to-digital converter (**ADC**). An STM32 device has the following advantages: structured architecture, the design time is less (just configure and program), higher precision (depending on the design) than analog devices, and lower power consumption, but it has some drawbacks such as a closed architecture and sometimes a lower speed than an FPGA. Finally, an FPGA has the following benefits: digital implementation, quick processing because of the clocks it can use, accuracy, and freedom of design/open architecture, i.e., the design can be reprogrammed and optimized. On the other hand, this device presents some disadvantages such as high current consumption

and longer design times than devices that already have blocks established. Because an FPGA can be reconfigurable, has a flexible design, and rapid prototyping has been helpful in implementing continuous-time and discrete-time chaotic dynamical systems better than other devices. In particular, FPGAs have been very useful to implement chaotic attractors in order to check mathematical models and observe their behavior in the real world [7] with the intention of obtaining applications based on chaos. For example, Tlelo et al. [8] implemented a chaotic dynamical system in an FPGA through an artificial neural network with the purpose of encrypting data, where random binary sequences are generated. Sundarapandian et al. [9] created a dynamical hyperchaotic system, as well as the realization of its circuit using MultiSim, and their implementation on FPGA for the application of a new encryption scheme in color images.

In relation to discrete-time systems, Aboulseoud and Ismail [10] reported on a floating-point hardware design of a fractional-order chaotic map and their implementation in an FPGA for image encryption. Thane and Chaudhari [11] proposed an FPGA-based cryptographically secured pseudo-random number generator (**PRNG**) by using a piecewise linear (**PWL**) map. Wang et al. [12] presented an FPGA implementation, as well as an analysis of discrete chaotic maps, which can be useful in applications. The use of some of these one-dimensional maps has been workable for applications that require high computation efficiency because of their low implementation costs [13,14]. In this regard, over the last decade, dynamical systems without fixed points or equilibrium points have been proposed. These systems have been the object of important consideration, since the mere fact of not having fixed points makes it difficult to determine their dynamics [15–18]. In the implementation of a PRNG, a system without a fixed point presents advantages over a system with a fixed point due to there being no orbit that converges to the fixed point for any initial condition. Additionally, if the initial condition is an eventually fixed point, then the orbit will be periodic for an $n$ iteration. Note that this could happen at the first iteration, the second iteration, and so on. It is worth mentioning that this does not occur in a system without a fixed point, so maps with more complex dynamics can be obtained and this fact is a primordial property in certain applications such as encryption schemes based on PRNGs. Since a map without a fixed point has no eventual fixed points, then the space of initial conditions leading to chaotic behavior is larger than a map with a fixed point. We have that the initial condition is part of the key space of an encryption scheme, then the key space increases the probability of generating a greater number of cryptographically secure sequences. For instance, Yu et al. [19] described an approach to obtain a PRNG by using a hyperchaotic system without an equilibrium point based on a memristor and its realization in FPGA. We remark that this last work shows a continuous dynamic system, this fact leads to employing some integration methods, for example, the Runge–Kutta algorithm, to solve and implement the dynamical system on the FPGA. When the integration methods used are simple, FPGA resources are reduced, but this results in larger rounding errors than if more complex integration methods were used. However, regardless of the method used, the problem of rounding errors will always be there. For a discrete dynamic system, an integration algorithm is unnecessary. Therefore, both the resources used and the rounding errors will be lower than if an integration method was applied (if a proper design is performed).

While there are other works where discrete systems are implemented in an FPGA in order to reproduce their chaotic dynamics, to the best of the authors' knowledge, no study has been conducted on implementing a one-dimensional map without fixed points on an FGPA. In view of the advantages stated in the previous paragraph, and that a PWL system is less difficult to implement compared to those built on nonlinear functions, in this work, we present an implementation of a PWL chaotic map without fixed points in an FPGA. In order to accomplish this task, and as mentioned in [20], in a first step, we make a computer simulation of the PWL map using floating-point arithmetic, specifically, we find, via Matlab, time series for different values of the parameters of the PWL map. As a the second step, we determine the range in which the values of the system lie and thus

establish the integer and the fractional part. As a third step, we carry out the design and simulation of the PWL map using fixed-point arithmetic (a dynamical system without fixed point refers to the fact that there is no $x$ such that $f(x) = x$, but a fixed point arithmetic refers to the computation of the value of $x$). In relation to this, we first show the design of the block diagram in which we synthesize all the operations that allow us to obtain the iterations of the PWL map. Second, we calculate the time series using fixed-point arithmetic for the same values of the PWL parameters that were used in the Matlab simulation, but now in the C language and in VHDL. As a final step, we perform the implementation of the PWL map design on the FPGA. The experimental results achieved with the FPGA implementation for particular values in the parameters by which the PWL map is chaotic show that these agree with the numerical results obtained with the C, VHDL, and Matlab languages, i.e., the same behavior can be observed in the time series produced with the experimental implementation and simulations.

In synthesis, the research conducted in this work contributes to the fact that new mappings without fixed points such as the one presented here can be taken into consideration in encryption schemes, in particular in the experimental discovery of PRNGs (through an electronic device such as an FPGA), and, therefore, application in various technologies that require an encryption system is possible. The rest of the work is presented as follows: in Section 2 we present the PWL map used in this work. In Section 3 we show an analysis of its dynamics. Section 4 contains the numerical simulations and the experimental realization of the PWL map in the FPGA. Finally, in Section 5 the conclusions of this research are given.

## 2. PWL Map without Fixed Points

In this section , we present the difference equation that defines the chaotic map used in this work, as well as a brief analysis of its dynamics. This map is a one-dimensional PWL mapping designed by García-Grimaldo and Campos Cantón [17], which, as will be seen later on, has no fixed points and exhibits chaotic dynamics when its parameters are within a certain range. Equation (1) describes the dynamic system based on a PWL map

$$x_{n+1} = \begin{cases} m_1 x_n + b_1 & \text{for} \quad -b_2 \leq x_n \leq -a, \\ m_2 x_n + b_2 & \text{for} \quad -a < x_n < 0, \\ m_2 x_n - b_2 & \text{for} \quad 0 \leq x_n < a, \\ m_1 x_n - b_1 & \text{for} \quad b_2 \geq x_n \geq a, \end{cases} \tag{1}$$

where $m_1 \neq 0$, $m_2 \neq 0$. The values $0 < a$ and $b_2$ are set as shown in (2) and (3).

$$a = \frac{b_1}{m_1}, \tag{2}$$

$$b_2 = \frac{m_2 \, b_1}{m_1}. \tag{3}$$

Note that $m_i$ and $b_i$ have the same sign, with $i = 1, 2$.

Fixed points. As we said previously, the PWL map given by (1) could present the absence of fixed points, specifically when its parameters $m_1$ and $m_2$ take determined values. In a formal way, García-Grimaldo and Campos-Cantón [17] establish a means of following Theorem 1, the hypothesis for which it is obtained a dynamical system without fixed points, as well as Theorem 2, which is proved to be chaos in Devaney's sense for particular values of (1).

**Theorem 1.** *Let $f$ be a PWL map given by (1), such that $m_2 > 0$ and $m_1 \in (-\infty, 0) \cup (0, 1)$, therefore, the PWL map (1) has not fixed point.*

**Theorem 2.** *The map given by* (1) *such that* $f : [-25, 25] \longmapsto [-25, 25)$, *with values* $m_1 = 0.80$, $m_2 = 5$, $b_1 = 4$, $b_2 = 25$, *has chaotic behavior in Devaney's sense.*

Figure 1 shows the graph of the map given in Theorem 2, from Theorem 1, with the values $m_1 = 0.8$, $m_2 = 5$, $b_1 = 4$, and $b_2 = 25$, the PWL map has no fixed points.

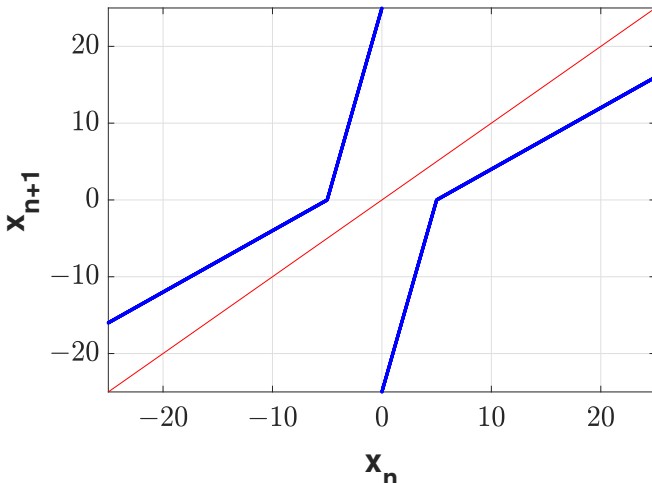

**Figure 1.** PWL map given by (1). $m_1 = 0.8$, $m_2 = 5$, $b_1 = 4$, and represented by the blue lines. The red line is the identity map.

### 3. Dynamical Analysis: Bifurcation Diagram and Lyapunov Exponents

In this part, we will perform an examination of the dynamics for the PWL map (1). First, we present the bifurcation diagram which will allow us to identify the regions in which there are non-periodic dynamics. Subsequently, we show the graph which displays the Lyapunov exponents obtained by varying the parameter $m_1$.

In Figure 2a we can see the bifurcation diagram which was generated by modifying the parameter $m_1$ in the closed interval $[0.1, 1]$, in addition, for each of the values of $m_1$, the initial condition was taken as $x_0 = 0.11$. The values $m_2$ and $b_1$ were set to 5 and 4, respectively. As can be noted when $m_1$ lies in the intervals $(0.1, 0.2)$ and $(0.25, 0.45)$, the orbit generated from the successive iterations of map (1) is periodic, being of period four in the first interval and of period six in the second interval. The other way, there are two intervals in the parameter $m_1$ where there are chaotic dynamics: $0.2 < m_1 < 0.25$ and $0.45 < m_1 \leq 1$. From the aforementioned, we can notice that this PWL system, presents a route to chaos different from other maps, such as the logistic one in which, as it is well known, presents a route to chaos by means of period doubling, while for map (1), it starts first with a periodic behavior and immediately when $m_1$ reaches a certain value it goes to a dynamic with chaotic behavior, which is maintained until the value of $m_1$ meets a certain value, after that, a periodic dynamic is suddenly produced and eventually a chaotic behavior emerges once again.

In order to determine and corroborate what was observed in the previous bifurcation diagram, where periodic and chaotic dynamics were detected in certain intervals, the results of the calculation of the Lyapunov exponents are shown below by means of the following definition:

$$\lambda(x_0) = \lim_{n \to \infty} \frac{1}{n} \sum_{j=0}^{n-1} \ln |f'(x(j))|. \tag{4}$$

where $\lambda(x_0)$ is the Lyapunov exponent calculated for a given initial condition $x_0$. A value of $\lambda > 0$ as well as the orbit of $x_0$ is bounded, implying the existence of chaos.

Figure 2b displays the calculation of the Lyapunov exponents of map (1) by taking values of $m_1$ in the closed interval $[0, 1]$, $m_2 = 5$, $b_1 = 4$ and the initial condition $x_0 = 0.11$. As can be corroborated in this plot, the Lyapunov exponent is non-positive in the same

intervals where periodic dynamics were obtained in the bifurcation diagram, i.e., in the intervals where $0 < m_1 < 0.2$ and $0.25 < m_1 < 0.45$, and is positive for the intervals: $0.20 < m_1 < 0.25$ and $0.45 < m_1 < 1$. Given that the orbits in each of these two intervals are bounded and $\lambda > 0$, then chaos is present in each of them.

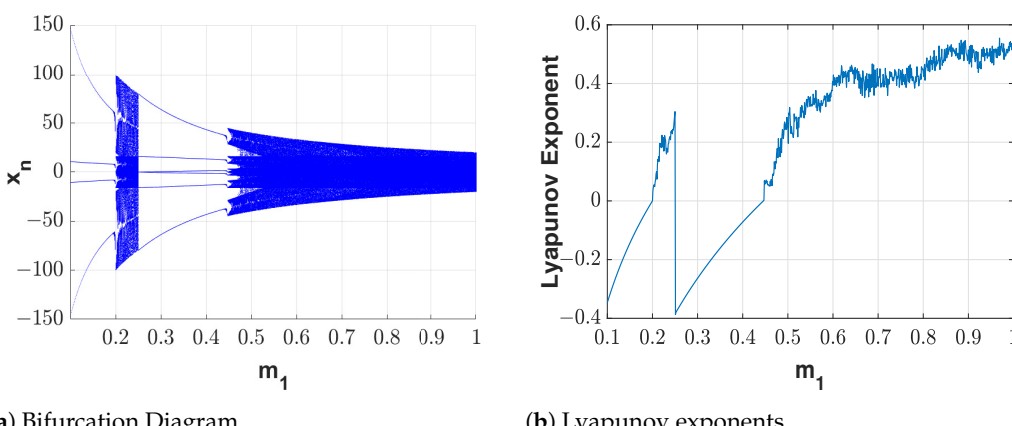

(**a**) Bifurcation Diagram.　　　　　　　　(**b**) Lyapunov exponents.

**Figure 2.** Dynamical analysis of the PWL map (1) by taking values of $m_1$ in the closed interval $[0, 1]$, $m_2 = 5$, $b_1 = 4$ and the initial condition $x_0 = 0.11$.

## 4. FPGA Implementation

As is well known, FPGA devices have recently been used for the electronic implementation of dynamical systems in continuous time; however, one of the disadvantages of employing such systems is that numerical methods must be applied to discretize the system, which leads to the consumption of more resources of the FPGA. On the other hand, the FPGA implementation of discrete-time dynamic systems offers the advantage that the aforementioned discretization step is not required due to the nature of the system. In this work, the implementation of a discrete PWL map on an FPGA is carried out and the fact that it is built by means of simple functions leads to lower utilization of FPGA resources.

In Figure 2b, it is possible to see a positive Lyapunov exponent for a wide range of values of the parameter $m_1$. This implies chaotic dynamics for a wide range of values of $m_1$, and the parameters $m_2 = 5$ and $b_1 = 4$. Then there are many options to select the value of the parameter $m_1$ for the FPGA implementation of the PWL map (1), that is, the value of the parameter $m_1$ can be chosen arbitrarily considering only when the Lyapunov exponent is positive. A wide range of values of $m_1$ of the PWL map (1) is an advantage in generating an encryption scheme because the parameter values are part of the key space of the system. With respect to other maps such as the logistic map, if the parameter value is always taken to be $\mu = 4$, then it is not considered as part of the key when used in encryption schemes.

In this work, three different sets of parameters are used to carry out the simulations in Matlab and C. Finally, one of these three sets of parameters for the PWL maps is used for implementation in VHDL and on the FPGA, and we select the set of parameters that exhibits chaos in the Devaney sense to show that it is feasible to reproduce chaos experimentally in a mapping that exhibits chaos theoretically.

### 4.1. Matlab Simulation

Before performing the simulation of the PWL map in VHDL, as well as its implementation on the FPGA, the time series of the PWL map computed with Matlab for three different sets of values are shown below to compare them with the time series computed in C, VHDL and experimentally with the FPGA. Additionally, the histograms corresponding to each map are displayed to observe how their distribution changes when one of their parameters is varied.

For Matlab, the data test volume was 10 0000 iterations for each set of parameters shown in Figure 3 and the same initial condition $x_0 = 0.1$ (Matlab makes the calculations using floating point arithmetic according to the standards of the IEEE). In Figure 3, we can

appreciate the time series and histograms for the three different maps. We can observe that as in Figure 2, there is a chaotic behavior for the values present in Figure 3c,e, while for the parameters in Figure 3a, there is periodic behavior. Furthermore, the striped structure obtained in Figure 3c occurs because there are bars that are close together with a frequency close to each other, as shown in the histogram in Figure 3d. As the value of $m_1$ increases, the distribution of values is concentrated closer to $x_n = 0$ and the stripe width is thinner (see Figure 3e,f).

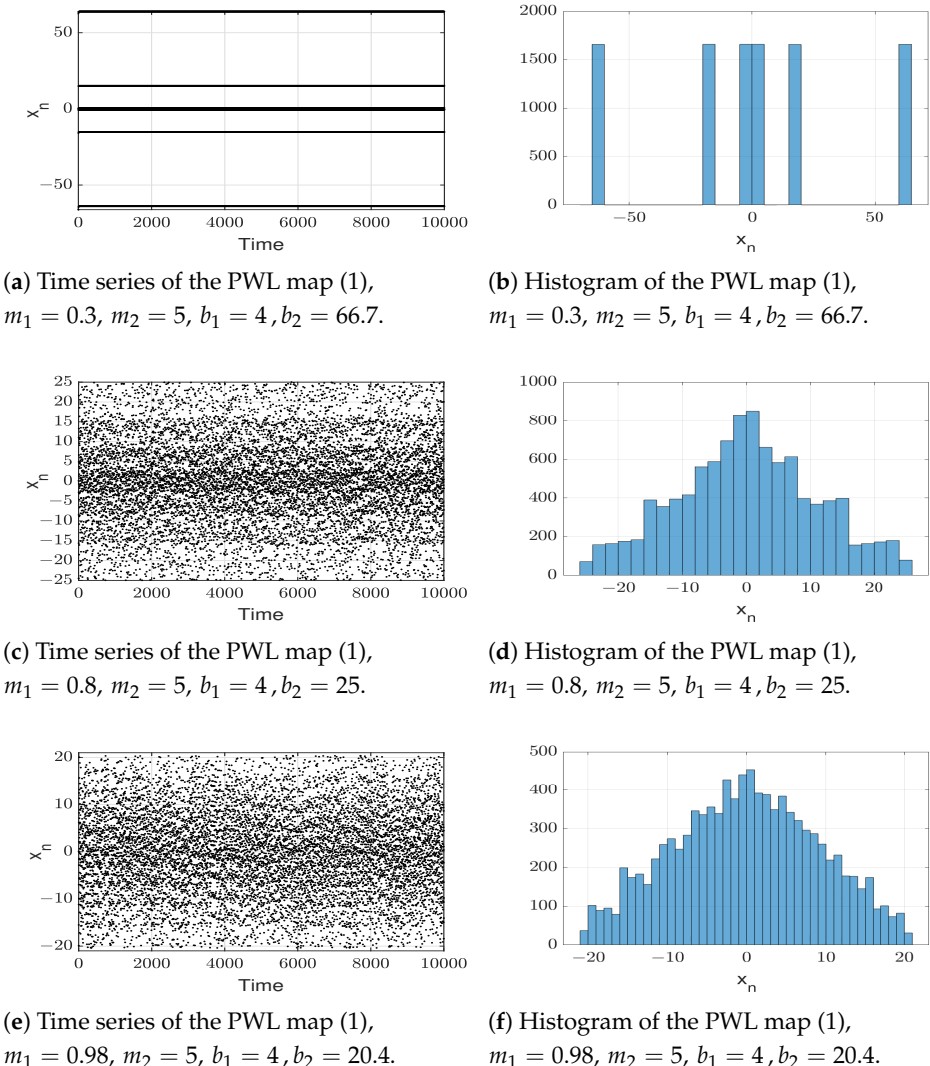

(**a**) Time series of the PWL map (1), $m_1 = 0.3$, $m_2 = 5$, $b_1 = 4$, $b_2 = 66.7$.

(**b**) Histogram of the PWL map (1), $m_1 = 0.3$, $m_2 = 5$, $b_1 = 4$, $b_2 = 66.7$.

(**c**) Time series of the PWL map (1), $m_1 = 0.8$, $m_2 = 5$, $b_1 = 4$, $b_2 = 25$.

(**d**) Histogram of the PWL map (1), $m_1 = 0.8$, $m_2 = 5$, $b_1 = 4$, $b_2 = 25$.

(**e**) Time series of the PWL map (1), $m_1 = 0.98$, $m_2 = 5$, $b_1 = 4$, $b_2 = 20.4$.

(**f**) Histogram of the PWL map (1), $m_1 = 0.98$, $m_2 = 5$, $b_1 = 4$, $b_2 = 20.4$.

**Figure 3.** Various time series and corresponding histograms for the PWL map (1).

### 4.2. Arithmetic Operations

In digital electronic devices, two formats can represent numerical values: fixed-point and floating-point arithmetic. In fixed-point notation, a number $n$, the binary point separating its integer and fractional part, is limited to a particular or fixed position in the bit pattern. In floating-point notation, a number $n$ is expressed in the form $n = m2^e$, where $m$ is the mantissa and $e$ is the exponent.

To implement dynamic systems in an FPGA, it is advisable that the calculation of the operations be performed using fixed-point arithmetic instead of floating-point ones since in this format the values can be computed more quickly, and the use of hardware resources and costs is minor. Moreover, if the dynamic system is not represented by highly complex functions, as of the PWL maps used in this work, the use of floating-point arithmetic can adequately represent the dynamics of the system. Because of this, in this article, we decided to use the arithmetic of fixed point to compute the PWL system (1).

To establish within this arithmetic an adequate representation of the mathematical model of the PWL map, it is necessary to determine: the range that the integer values of the parameters can take, the values obtained by performing the operations describing the PWL map (1), and the amplitude of the state variable $x$. Given the time series of Figure 3 and the bifurcation diagram of Figure 2b, we can note that the values of the iterations are lower and upper bounded by $-b_2$ and $b_2$, respectively. From this, by Theorem 3 we established the conditions that guarantee that all the operations involved in the PWL map, its parameters, as well as the amplitude state variable are bounded in the closed interval $[-b_2, b_2]$.

**Theorem 3.** *Let f be a PWL map defined as* (1), *such that* $b_1 > 1, m_2 > 1$ *and* $m_1 \in (0, 1)$, *then* $m_1$, $m_2$, $b_1$, $m_1 x_n$, $m_2 x_n$, $m_1 x_n + b_1$, $m_1 x_n - b_1$, $m_2 x_n + b_2$, *and* $m_2 x_n - b_2$ *are bounded on the closed interval* $[-b_2, b_2]$.

**Proof.** First, we prove that $m_1, m_2, b_1 \in [-b_2, b_2]$. By hypotheses, $b_1 > 1, m_2 > 1$, then $b_1 > m_1$, thus $\dfrac{b_1}{m_1} > 1$, since $b_2 = \dfrac{m_2 b_1}{m_1}$, then, $-b_2 < m_2 < b_2$. In analogous way, it can proved that $m_2, b_1 \in [-b_2, b_2]$.

Second, we prove that $m_1 x_n, m_2 x_n \in [-b_2, b_2]$. If $-b_2 \leq x_n \leq \dfrac{-b_1}{m_1}$, and by hypotheses $m_1 \in (0, 1)$, then $-b_2 \leq x_n < m_1 x_n < 0 < b_2$.
If $\dfrac{-b_1}{m_1} < x_n < 0 < \dfrac{b_1}{m_1}$, then $-b_2 = \dfrac{-m_2 b_1}{m_1} < m_2 x_n < 0 < \dfrac{m_2 b_1}{m_1} = b_2$.

Third, we prove that $m_1 x_n + b_1, m_1 x_n - b_1, m_2 x_n + b_2, m_2 x_n - b_2$. If $-b_2 \leq x_n \leq \dfrac{-b_1}{m_1}$ and the fact that $b_1 > 1$ then $-b_2 < m_1 x < m_1 x_n + b_1 \leq 0 < b_2$. If $\dfrac{-b_1}{m_1} < x_n < 0 \dfrac{b_1}{m_1}$ and due to $-b_2 = \dfrac{-m_2 b_1}{m_1} < m_2 x_n < 0$, then $0 < m_2 x_n + b_2 < b_2$. In an analogous way, it can be proven that $m_1 x_n - b_1, m_2 x_n - b_2$. □

Under the conditions of Theorem 3, we can state that the range that the integer part can take for all values is from $-\overline{b_2}$ to $-\overline{b_2}$, where $-\overline{b_2}$ is the next integer greater than $b_2$. In this work, for the C and VHDL simulations, we will use three common architectures: 16, 32, and 64 bits. Thus, once the value of $b_2$ is defined, we will determine the number of bits required to represent all integer values in the range $[-\overline{b_2}, \overline{b_2}]$ or vice versa, we can give the number of bits for the integer part and determine the range, and as a result know the numbers of bits that will be available for the fractional part in each of the above architectures.

### 4.3. Block Diagram of the PWL Map

After determining the arithmetic computation, the next step is to identify the type of blocks that will be required for the simulation and implementation of the PWL map in the FPGA. Figure 4 displays the block diagram used for the implementation of the PWL map that is defined in four subintervals $I_i$, $i = 1, \ldots, 4$ as we can see in (1). Figure 4 shows the adder, subtractor, and multiplier blocks needed to perform the arithmetic operations. Additionally, since we have a piecewise map, it is required to determine in each $n$-th iteration in which interval $I_i$, with $i = 1, \ldots, 4$, $x_n$ is located. In Figure 4, it can be seen that this could be determined with two blocks: the comparator block, which precisely has as input signals the signal $x_n$ and the threshold signal $a$, and the encoder block, in which the output signal "SEL2" allows us to determine the correct value of $x_{n+1}$, which is represented by the output signal $f_r$ of the multiplexer with input signals "SEL2" and $f_1, f_2, f_3, f_4$. Finally, the signal $f_c$ is fed to a register block which allows us to store each of the computed iterations. As can be seen in Figure 4, there is also a read-only memory (**ROM**) block in which the values of the parameters that remain constant are stored. In addition, in order to start and end the iterative process, a state control machine was designed, which is composed of the following blocks: a multiplexer block, which allows us to choose between

the initial condition $x_0$ or the condition $x_{n+1}$, and a control unit (**CU**) block that allows to start and end the computation of the PWL map dynamics.

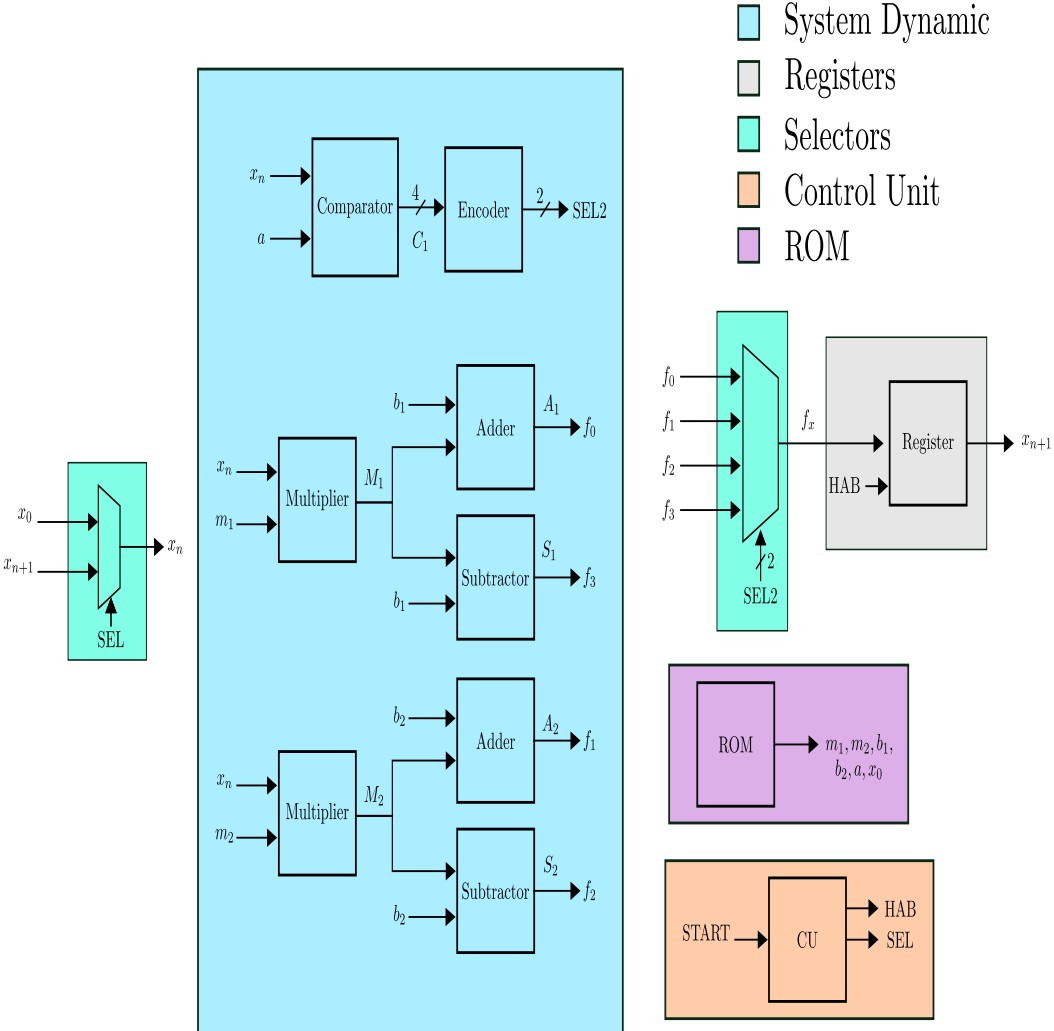

**Figure 4.** Block diagram of the PWL map (1).

### 4.4. Simulation in C and VHDL

Usually, once the arithmetic computation is determined, as well as the design of the blocks, the next step is to create the code in a description language such as VHDL to carry out the simulation process. However, in this work, we also present simulations derived from the creation of a code in the C language, in which the arithmetic computation was done using fixed-point arithmetic. The main reason to perform the implementation in C is intending to detect design errors before the simulation in VHDL and its subsequent implementation in the FPGA, where it is more difficult to detect such errors. We use the C programming language with a GCC compiler on a UNIX base operating system, in the code we use the following data type: double-8 bytes, int128-16 bytes, long-8 bytes, and int-4 bytes.

In each of the three architectures: 16, 32, and 64 bits, we occupy eight bits for the integer part. Therefore, the range of values that the integer part can take is $[-2^8 - 1, 2^8 - 1]$, this is, with eight bits it is possible to represent PWL maps defined by (1) where the amplitude, parameters, and dynamics belong to the close interval $= [-\overline{b} = -255, \overline{b} = 255]$, the reason for this was that the simulations made with of the time series of the three maps described above had a range which could avoid falling into an overflow; however, it is important to

mention that it can take a larger range based on the conditions of Theorem 3. Table 1 lists the number of bits used in each of the three architectures.

**Table 1.** Number of bits in each architecture.

| Architecture | # Sign-Bit | # Bit-Integer | # Bit-Fractional |
|---|---|---|---|
| 16 bits | 1 | 8 | 7 |
| 32 bits | 1 | 8 | 23 |
| 64 bits | 1 | 8 | 55 |

Figure 5 displays the time series (TS) for the same three PWL employed on simulation made with Matlab but now obtained by simulating it in the C language using fixed-point arithmetic with three distinct architectures. For the PWL simulation using C language, we also calculate 100,000 iterations, and an initial condition $x_0 = 0.1$. For the three maps it was taken $m_2 = 5$, $b_1 = 4$; for PWL-1, $m_1 = 0.3$, $b_2 = 66.7$; for PWL-2, $m_1 = 0.8$, $b_2 = 25$; for PWL-3, $m_1 = 0.98$, $b_2 = 20.4$.

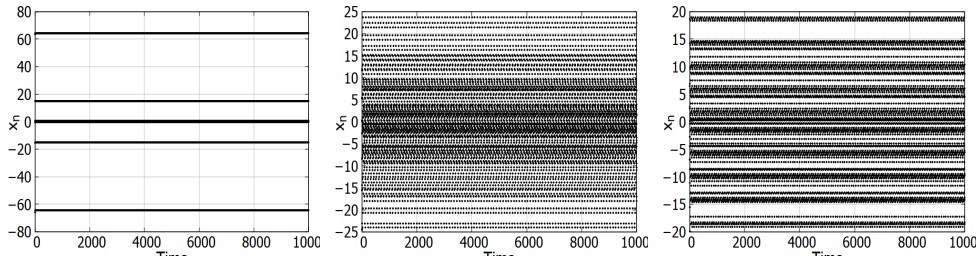

(**a**) TS of PWL-1 using 16 bits. (**b**) TS of PWL-2 using 16 bits. (**c**) TS of PWL-3 using 16 bits.

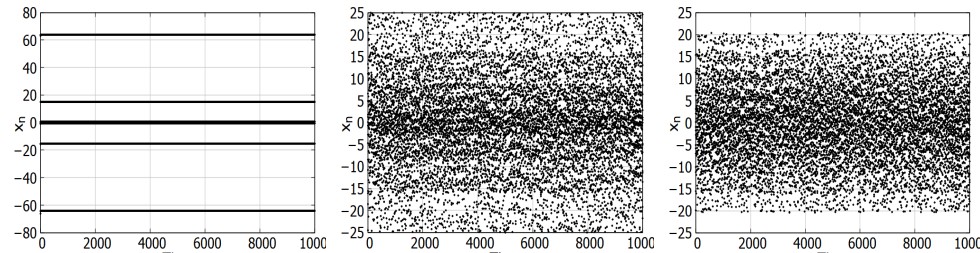

(**d**) TS of PWL-1 using 32 bits. (**e**) TS of PWL-2 using 32 bits. (**f**) TS of PWL-3 using 32 bits.

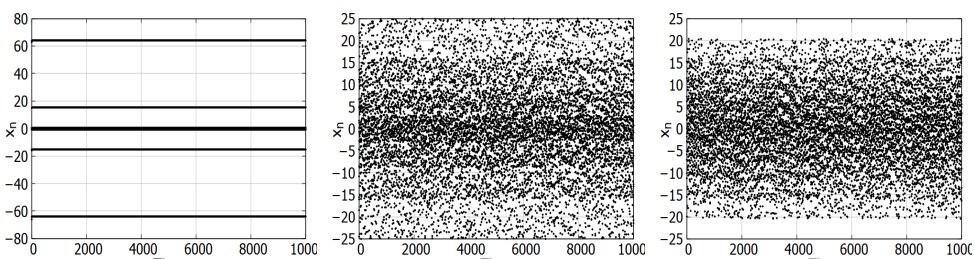

(**g**) TS of PWL-1 using 64 bits. (**h**) TS of PWL-2 using 64 bits. (**i**) TS of PWL-3 using 64 bits.

**Figure 5.** Time series for three distinct PWL maps with three different architectures.

In Figure 5b,c, we can detect that with a 16-bit architecture, there are discrepancies in the time series of map PWL-1 and map PWL-2 regarding the series of the same maps simulated with Matlab, which can be seen in Figure 3c,e, respectively. However, for the three PWL maps the time series in Figure 5d–i with the architectures of 32 and 64 bits, respectively, shows the same dynamical behavior that was exhibited in the time series in Figure 3a,c,e. Thus, C language simulations of maps using 32-bit and 64-bit fixed-point

arithmetic show that it is achievable to reproduce the dynamics now in VHDL and in the FPGA.

From the simulations made in the C language, it was possible to reproduce the dynamics of the PWL maps defined by (1) using 32-bit and 64-bit architectures for three sets of parameters. In this part, we display the VHDL simulation of the PWL map. We select the set of parameters to generate chaos, as already detailed in previous sections, chaotic behavior was demonstrated through Devaney's definition, as well as by using Lyapunov exponents. For this reason, we consider it important to carry out its implementation experimentally, since the fact of achieving it shows a map without fixed points with the presence of chaotic dynamics in three senses: theoretical, numerical simulation, and experimentation.

For the PWL map simulation using VHDL, we also compute 10,000 iterations and the parameters are $m_1 = 0.8$, $m_2 = 5$, $b_1 = 4$, $a = 5$, $x_0 = 0.1$. To implement the PWL map, each of the blocks present in the diagram in Figure 4 is programmed with VHDL using 64-bit fixed-point architecture, given that this has higher precision than the 32-bit architecture for the calculation of each of its iterations.

Figure 6 shows the values of the first iterations of the PWL map (1) obtained by the VHDL simulation. Since these values are in hexadecimal format, we also decided to calculate the same number of iterations using C language with the same architecture and in hexadecimal format to compare both results and determine if there were any discrepancies between the values obtained with VHDL and with C.

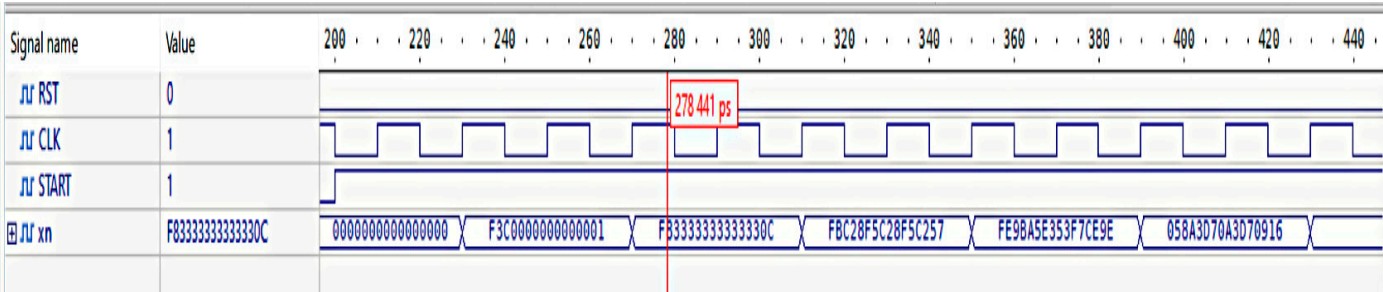

**Figure 6.** VHDL simulation of the PWL map (1). $m_1 = 0.8$, $m_2 = 5$, $b_1 = 4$, $a = 5$, $x_0 = 0.1$.

Table 2 presents the values in a hexadecimal format of the first 10 iterations and the last iterations in both languages. As we can see, exactly the same results are achieved in both the first and last iterations in C-hexadecimal and in VHDL. Therefore, the analysis, design, programming, and simulation of the block diagram representing the PWL map (1) are adequate. Thus, the last step to be carried out is the synthesis and implementation of the map in the FPGA.

*4.5. FPGA Implementation*

Figure 7 shows the experimental implementation of the PWL map. Figure 7a depicts the necessary devices to obtain the PWL time signal, which consists of an FPGA Basys 3 Xilinx Artix-7 XC7A35T-1CPG236C, a linear power supply at 5 V, a 16-bit DAC, and an oscilloscope to visualize the output signal, which can be seen in Figure 7b. Here, in the time series reproduced on the oscilloscope, we can detect the same type of chaotic behavior as that exhibited in the simulations done in Matlab, C, and VHDL. In each of these simulations, as well as in the FPGA implementation, the same lines are observed, i.e., there is the same type of distribution. In this way, it has been possible to reproduce experimentally the chaotic dynamics of the PWL map (1). Finally, Table 3 shows the resources used by the FPGA Basys 3 Xilinx Artix-7 XC7A35T-ICPG236C to implement the PWL map (1). These data were acquired when the FPGA is operated with a clock signal of 100 MHz.

**Table 2.** Iterations of the PWL map.

| Iteration | VHDL-Value | C-Hexadecimal Value |
|---|---|---|
| 1 | f3c0000000000001 | f3c0000000000001 |
| 2 | f83333333333330c | f83333333333330c |
| 3 | fbc28f5c28f5c257 | fbc28f5c28f5c257 |
| 4 | fe9ba5e353f7ce9e | fe9ba5e353f7ce9e |
| 5 | 058a3d70a3d70916 | 058a3d70a3d70916 |
| 6 | 026e978d4fdf3a89 | 026e978d4fdf3a89 |
| 7 | ffa8f5c28f5c24ad | ffa8f5c28f5c24ad |
| 8 | 0acccccccccccb761 | 0acccccccccccb761 |
| 9 | 06a3d70a3d7092d6 | 06a3d70a3d7092d6 |
| 10 | 034fdf3b645a0f26 | 034fdf3b645a0f26 |
| 9996 | 0a3c7dcc356a64ca | 0a3c7dcc356a64ca |
| 9997 | 063064a35deeb728 | 063064a35deeb728 |
| 9998 | 02f383b5e4bef900 | 02f383b5e4bef900 |
| 9999 | 005c695e5098c73c | 005c695e5098c73c |
| 10,000 | f54e0ed792fbe42c | f54e0ed792fbe42c |

**Table 3.** Hardware resources for the implementation of PWL system (1) by using the FPGA Basys 3 Xilinx Artix-7 XC7A35T-ICPG236C.

| Resources | PWL | Available |
|---|---|---|
| LUTs | 700 | 20,800 |
| FF | 127 | 41,600 |
| DSP | 24 | 90 |
| Multipliers | 2 | – |
| Adders | 2 | – |
| Subtractors | 2 | – |
| Comparators | 1 | – |
| Latency (ns) | 40 | 20 |

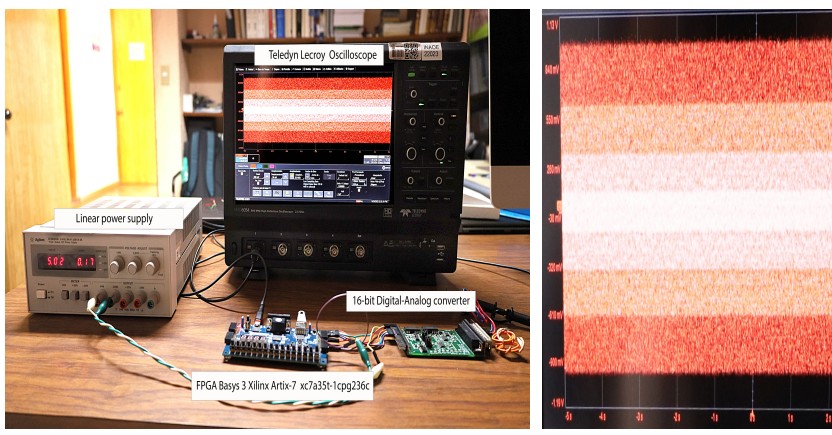

(**a**) Experimental setup.　　　　　　　　　　(**b**) Experimental time series.

**Figure 7.** FPGA implementation of the PWL map (1). $m_1 = 0.8$, $m_2 = 5$, $b_1 = 4$, $a = 5$, $x_0 = 0.1$.

## 5. Conclusions

In this work, the analysis, numerical simulation, and implementation in an FPGA of a PWL map without fixed points have been presented. This implementation is an alternative to encryption schemes because it has some advantages such as a wide range of values of the $m_1$ and this helps to increase the key space. To achieve the proposed objectives, a series of steps were carried out, firstly, the theoretical analysis was addressed with a couple of theorems on the difference equation that defines the PWL map. In the first, the conditions were established so that there are no fixed points, and in the second, a chaotic map in

the sense of Devaney was exhibited. Second, numerical simulations were performed to analyze the dynamics using bifurcation diagrams and Lyapunov exponents. The results indicated that the PWL map has continuous ranges where chaotic dynamics are developed, this characteristic evidences that its parameters can be used as part of the key of some encryption schemes, unlike other maps such as the logistic one, in which its parameter is generally not part of the key. Third, time series were shown by simulating them in Matlab using three sets of parameters of the PWL map. Fourth, it was determined that it was best suited for the calculation of the values of the PWL map iterations to use fixed-point arithmetic in the simulation process in C and VHDL, as well as for the implementation of the PWL map in an FPGA. In connection with this, we established a theorem that provided the conditions that determine the range in which the integer part of the values represented in the fixed-point notation of the PWL map is guaranteed to be bounded. Fifth, we presented the design of the block diagram in which the relationships between the operations that allow defining the PWL map were established. Sixth, we displayed the time series generated in the C language for the PWL map obtained from the simulation in Matlab using 16-, 32-, and 64-bit fixed point architectures. The results showed that, with the 16-bit architecture, the time series differ with respect to those obtained with Matlab, thus not correctly reproducing the dynamics of the PWL map. However, with the 32-bit and 64-bit architectures, the results revealed very similar time series with respect to those found with Matlab, with the 64-bit having a higher precision by using a greater number of bits for the fractional part. Based on this, we decided to use a 64-bit architecture for the simulation in VHDL, as well as in the implementation in the FPGA to the PWL map (the one that has been shown to have chaos in the sense of Devaney and through Lyapunov exponents). The results of the simulation in VHDL compared with those obtained in C in hexadecimal format showed that the same values were found in the iterations of the PWL map and, therefore, the chaotic behavior was reproduced, which proved that the design and programming of the PWL map in the VHDL language were correct, so as a last step the implementation of the PWL map in the FPGA was performed. The experimental results given showed that the time series exhibited the chaotic dynamics displayed in the series with Matlab and C. In this way, we could experimentally reproduce the chaotic dynamics of the PWL map without fixed points. Finally, from the results presented, the PWL map without fixed points employed in this work may be an option for application in encryption schemes or for the generation of cryptographically secure pseudo-random sequences.

**Author Contributions:** Conceptualization, C.G.-G., E.T.-C. and E.C.-C.; methodology, C.F.B.-M.; software, C.G.-G.; validation, C.F.B.-M. and E.T.-C.; formal analysis, C.G.-G., C.F.B.-M. and E.C.-C.; investigation,C.G.-G. and E.C.-C.; resources, E.T.-C. and E.C.-C.; data curation, C.G.-G. and E.C.-C.; writing—original draft preparation, C.G.-G. and E.C.-C.; writing—review and editing, E.T.-C; visualization, C.G.-G.; supervision, E.C.-C.; project administration, E.C.-C.; funding acquisition, E.C.-C. All authors have read and agreed to the published version of the manuscript.

**Funding:** This research was funded by CONACYT grant number A1-S-30433.

**Data Availability Statement:** The data used to support the findings of this study are included within the article.

**Acknowledgments:** C. García-Grimaldo is thankful to CONACYT for the scholarships granted. Eric Campos-Cantón acknowledges CONACYT for their financial support through Project No. A1-S-30433.

**Conflicts of Interest:** The authors declare that there are no conflicts of interest regarding the publication of this work.

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
