# Peer review of "FPGA Implementation of a Chaotic Map with No Fixed Point"

_electronics, doi:10.3390/electronics12020444_

Round 1

Reviewer 1 Report

Several drawbacks should be corrected, in order to publish the manuscript in “Electronics”

11.       Manuscript contains literature review of various variants of electronic devices with chaotic output signals, but there are no discussion of methods and criteria to compare the quality of pseudo random data for these systems, and finally to appreciate advantages of the new variant. As the authors has made the conclusion in sec.4 concerning cryptographically  secure sequences and encryption schemes based on the used PWL map, there should be some commentaries to compare the level and merits  of such schemes with some other RNG systems.

2.     Authors has paid a lot of attention to calculations with fixed  and floating points in sec. 1, 3, 3.2, and 4,  but commentaries are not   clear enough, if there were chances to see discrepancy for them. 

3.  Description of PWL map in sec.2 gives parameters of the used map, but the way to choose curves and their parameters for practical work is not disclosed.

4. Obtained structure of stripes in figs.4. and 6 for time series is preferably to be commented. How  it depends on PWL model parameters and with what data can be compared ?

5. Please, comment, the volume of testing for data given in fig.4 c). 

6. Check, please, in sec.3.1 the numbers of commented figures.

Author Response

Comments

  1. Manuscript contains literature review of various variants of electronic devices with chaotic output signals, but there are no discussion of methods and criteria to compare the quality of pseudo random data for these systems, and finally to appreciate advantages of the new variant. As the authors has made the conclusion in sec.4 concerning cryptographically secure sequences and encryption schemes based on the used PWL map, there should be some commentaries to compare the level and merits of such schemes with some other RNG systems.

Answer.

In the Introduction section, content has been added (highlighting the text in blue color) about the advantages as well as the contribution of implementing the proposed fixed-point-free map on the FPGA device.  In addition, the Conclusions section has been rewritten to better explain the contributions of this work. Originally marked as section 4, due to the changes made, it is now section 5 in this revised version.

Comment

  1. Authors has paid a lot of attention to calculations with fixed and floating points in sec. 1, 3, 3.2, and 4, but commentaries are not clear enough, if there were chances to see discrepancy for them.

Answer.

Thank you for your observation.  To explain in more detail the calculations in both arithmetics, we have added more information, calculations and theoretical aspects that can be found in the subsections (highlighting the text in blue color): 4.1. Matlab Simulation, 4.4 Simulation in C and VHDL and Section 5. In addition, a new subsection marked 4.2. Arithmetic operations, where it is explained how the use of fixed-point arithmetic was determined and justified in the C and VHDL simulations as well as in the FPGA implementation.

Note: Section 3 of the first version is now Section 4 of the revised version; subsection 3.2 has now been split into subsections 4.2 and 4.4. And Section 4 is now section 5.

Comment

  1. Description of PWL map in sec.2 gives parameters of the used map, but the way to choose curves and their parameters for practical work is not disclosed.

Answer

 At the beginning of section 4, before subsection 4.1 (highlighting the text in blue color), an explanation is given as to why the selected parameters were chosen. However, it is also worth noting that in that part an explanation is given as to why other parameters could be chosen. In addition, time series of two other sets of parameters of the PWL map, obtained through the same simulation process done previously, have been added with the intention of justifying what has been explained, as well as to make comparisons to give more support to this work.

Comment

  1. Obtained structure of stripes in figs.4. and 6 for time series is preferably to be commented. How it depends on PWL model parameters and with what data can be compared?

Answer

In subsection 4.1. Matlab, as well as in section 4.5 FPGA implementation, comments on the structure of the obtained time series strips are given 1. In addition, time series of a couple of parameter sets of the PWL map have been added to observe the dependence of PWL on the parameters.

Comment

  1. Please, comment, the volume of testing for data given in fig.4 c). 

Answer

On subsection 4.4, we added information of the volume of testing for data given in fig.4 c). The information given in such figure, now are present in Table 2 with the intention of a better understanding.

Comment

  1. Check, please, in sec.3.1 the numbers of commented figures.

Answer

Thank you for your observation. This issue has been corrected.

Reviewer 2 Report

The article under review is devoted to FPGA implementation a chaotic map without fixed points. In this study authors carried out the implementation of a discrete chaotic dynamical system without fixed points in a Xilinx FGPA device to be able to reproduce its chaotic behavior. The authors provide the bifurcation analysis and calculation of the Lyapunov exponent to prove the chaotic behavior. Also, the authors provide experimental results on FPGA and the numerical simulation of a chaotic map under investigation using Matlab and C implementation and experimental. In general, this manuscript is well-written, however, several shortcomings should be addressed:

  1. Please add a contribution at the end of the introduction section.
  2. Please clearly emphasizes what advantages you get by implementing a chaotic map without fixed-point arithmetic.
  3. For me, it’s not clear why numerical simulation is provided only in floating and fixed point arithmetic. Since the experimental implementation was performed using without fixed-points arithmetic, then the numerical simulation should also be performed in the same arithmetic.
  4. The description given in the 3.2 and 3.3 subsections is too vague. The description is brief and not clear.
  5. Figures 2a and 2b I recommend performing in the same style
  6. Figure 3 needs to be enlarged for better readability.
  7. Figures 4a and 4b should be made the same size.

Author Response

 Comments

  1. Please add a contribution at the end of the introduction section.

  1. Please clearly emphasizes what advantages you get by implementing a chaotic map without fixed-point arithmetic.   

  1. For me, it’s not clear why numerical simulation is provided only in floating- and fixed-point arithmetic. Since the experimental implementation was performed using without fixed-points arithmetic, then the numerical simulation should also be performed in the same arithmetic.

Note. First, we would like to apologize for any confusion that may have arisen around the concepts of "map without fixed points" and "fixed point arithmetic".

We want to clarify that a dynamical system with no fixed point refers to the existence of  such that , but a fixed-point arithmetic refers to the computation of the values of

Answers to Comments 1,2 and 3

 In the Introduction section, information has been added regarding the advantages as well as the contribution of implementing the proposed fixed-point-free map on the FPGA device.  In addition, the Conclusions section has been rewritten to further explain the contributions of this work. Originally marked as section 4, due to the changes made, it is now section 5.

Remark. Section 3 is now section 4; subsection 3.2 has now been split into subsections 4.2 and 4.4. Section 4 is now section 5.

Comment

  1. The description given in the 3.2 and 3.3 subsections is too vague. The description is brief and not clear.

 Answer

The description given in the 3.2 and 3.3 subsections are now given with more detail in 4.1, 4.2 and 4.4 subsections. 

In order to explain in more detail, the calculations made in arithmetic of floating point and arithmetic of fixed point, more information, calculations and theoretical aspects have been added, which can be found in the sections: 4.1. Matlab Simulation, 4.4 Simulation in C and VHDL and Section 5. In addition, a new subsection marked 4.2. Arithmetic operations, where it is explained how the use of fixed-point arithmetic was determined and justified in the C and VHDL simulations as well as in the FPGA implementation.

Comment

  1. Figures 2a and 2b I recommend performing in the same style

Answer

Thank you for your observation. This issue has been corrected.

Comment

  1. Figure 3 needs to be enlarged for better readability.

Answer

Thank you for your observation. Figure 3 has been enlarged. In this new version, Figure 3 is now Figure 4.

Comment

  1. Figures 4a and 4b should be made the same size.

 Answer

Thank you for your observation. In this new version, new figures 5a-i have been added, which are related to Figures 4a and 4b, and which have been made the same size.

Reviewer 3 Report

The issue that is addressed in the article is important and really topical. Random number generation - a high-quality sequence of random number sequences plays an important role in modeling systems, for performing simulation experiments to predict events or behavior of the modeled system and the original object.

There are a number of abbreviations in the article and not all of them are adequately explained. This may not cause any difficulties for an expert to understand the content of the article, but the common user and reader definitely needs a more detailed explanation of the mentioned methods. It would also be appropriate to pay more attention to the computer implementation of data types that are used in the program tools, programming languages and programming environments. Also, display accuracy is very important for "Real" data types and also the range for integer types when generating random numbers. It would be appropriate to elaborate the conclusion in more detail, to better justify the results. It is also important to show, or to theoretically justify the influence of input parameters on the quality and properties of a random number sequence. I positively evaluate the hardware solution for obtaining random numbers.

On page 5, line 144, "Figure 4b" should read "Figure 4c". The text under figure 4 (c) "(c) Values of the first iterations on C-hexadecimal" should be written in one line.

Author Response

Comment

  1. The issue that is addressed in the article is important and topical. Random number generation - a high-quality sequence of random number sequences plays an important role in modeling systems, for performing simulation experiments to predict events or behavior of the modeled system and the original object. There are a number of abbreviations in the article and not all of them are adequately explained. This may not cause any difficulties for an expert to understand the content of the article, but the common user and reader definitely needs a more detailed explanation of the mentioned methods.

Answer

Thank you for your observation. All abbreviations present in the text have been defined.

Comment

  1. It would also be appropriate to pay more attention to the computer implementation of data types that are used in the program tools, programming languages and programming environments. Also, display accuracy is very important for "Real" data types and also the range for integer types when generating random numbers

Answer

In order to explain in more detail, the calculations made in arithmetic of floating point and arithmetic of fixed point as well as the computer implementation of data types that are used in the program tools, programming languages, and accuracy, we have added more information, calculations and theoretical, which can be found in the subsections: 4.1. Matlab Simulation, 4.4 Simulation in C and VHDL and Section 5. In addition, a new subsection marked 4.2. Arithmetic operations, where it is explained how the use of fixed-point arithmetic was determined and justified in the C and VHDL simulations as well as in the FPGA implementation.

Remark. Section 3 is now section 4; subsection 3.2 has now been split into subsections 4.2 and 4.4. Section 4 is now section 5.

Comment

  1. It would be appropriate to elaborate the conclusion in more detail, to better justify the results.

Answer

The Conclusions section has been rewritten to further explain the contributions of this work. Originally marked as section 4, due to the changes made, it is now section 5.

In addition, in the Introduction section, content has been added (highlighting the text in blue color) regarding the advantages as well as the contribution of implementing the proposed fixed-point-free map on the FPGA device.

Comment

  1. It is also important to show, or to theoretically justify the influence of input parameters on the quality and properties of a random number sequence. I positively evaluate the hardware solution for obtaining random numbers.

Answer

In relation to this, at the beginning of section 4, before subsection 4.1 (highlighting the text in blue color), an explanation is given as to why the selected parameters were chosen. However, it is also worth noting that in that part an explanation is given as to why other parameters could be chosen. In addition, time series of two other sets of parameters of the PWL map, obtained through the same simulation process done previously, have been added with the intention of justifying what has been explained, as well as to make comparisons to give more support to this work and future applications on PRNGs.

Comment

On page 5, line 144, "Figure 4b" should read "Figure 4c". The text under figure 4 (c) "(c) Values of the first iterations on C-hexadecimal" should be written in one line

Answer

Thank you for your observation. This issue has been corrected.

Reviewer 4 Report

The authors present on the implementation of a piecewise-linear (PWL) chaotic map without fixed points on FPGA and show the usefulness  by numerical calculations.

However, there exist works on implementations of chaotic map for some applications as authors referred in this paper. One of Originalities of the authors' scheme is that they use chaotic maps without fixed points.The Authors should give explanation about advantages of using the chaotic map without fixed points.

Furthermore, this paper shows the validity of FPGA implementations of the PWL map (eq. (1) in this paper) with only one parameter set (m_1=0.8, m_2=5, b_1=4, a=5). The authors should show the validity by using other parameter sets and mention the optimal parameter set for the implementation.

I defer my recommendation until after I see the revision based on above comments.

Author Response

Comment

The authors present on the implementation of a piecewise-linear (PWL) chaotic map without fixed points on FPGA and show the usefulness by numerical calculations.

However, there exist works on implementations of chaotic map for some applications as authors referred in this paper. One of Originalities of the authors' scheme is that they use chaotic maps without fixed points. The Authors should give explanation about advantages of using the chaotic map without fixed points.

Answer

In the Introduction section, content has been added (highlighting the text in blue color) regarding the advantages as well as the contribution of implementing the proposed fixed-point-free map on the FPGA device.  In addition, the Conclusions section has been rewritten to further explain the contributions of this work. Originally marked as section 4, due to the changes made, it is now section 5.

Comment

Furthermore, this paper shows the validity of FPGA implementations of the PWL map (eq. (1) in this paper) with only one parameter set (m_1=0.8, m_2=5, b_1=4, a=5). The authors should show the validity by using other parameter sets and mention the optimal parameter set for the implementation.

Answer

In relation to this, at the beginning of section 4, before subsection 4.1 (highlighting the text in blue color), an explanation is given as to why the selected parameters were chosen. However, it is also worth noting that in that part an explanation is given as to why other parameters could be chosen. In addition, time series of two other sets of parameters of the PWL map, obtained through the same simulation process done previously, have been added with the intention of justifying what has been explained, as well as to make comparisons to give more support to this work and future applications on PRNGs.

Round 2

Reviewer 2 Report

Thanks to authors for careful improving the manusript.  I am satisfied with the latest version of manuscript and recommend publication.

Reviewer 4 Report

The authors have addressed my comments satisfactorily. I recommend this version for publication in this article.